# Reference Data for Diagnosis of Spondylolisthesis and Disc Space Narrowing Based on NHANES-II X-rays

**DOI:** 10.3390/bioengineering11040360

**Published:** 2024-04-08

**Authors:** John Hipp, Trevor Grieco, Patrick Newman, Vikas Patel, Charles Reitman

**Affiliations:** 1Medical Metrics, Houston, TX 77056, USA; tgrieco@medicalmetrics.com (T.G.); pnewman@medicalmetrics.com (P.N.); 2Anschutz Medical Campus, University of Colorado, Aurora, CO 80045, USA; vikas.patel@cuanschutz.edu; 3Medical University of South Carolina, Charleston, SC 29425, USA; reitman@musc.edu

**Keywords:** lumbar, spine, spondylolisthesis, intervertebral disc, radiographic, reference data

## Abstract

Robust reference data, representing a large and diverse population, are needed to objectively classify measurements of spondylolisthesis and disc space narrowing as normal or abnormal. The reference data should be open access to drive standardization across technology developers. The large collection of radiographs from the 2nd National Health and Nutrition Examination Survey was used to establish reference data. A pipeline of neural networks and coded logic was used to place landmarks on the corners of all vertebrae, and these landmarks were used to calculate multiple disc space metrics. Descriptive statistics for nine SPO and disc metrics were tabulated and used to identify normal discs, and data for only the normal discs were used to arrive at reference data. A spondylolisthesis index was developed that accounts for important variables. These reference data facilitate simplified and standardized reporting of multiple intervertebral disc metrics.

## 1. Introduction

Spondylolisthesis is a common clinical diagnosis and refers to the abnormal slip of one vertebra relative to an adjacent vertebra, typically assessed in the sagittal plane. Some degree of vertebral slip is normal in healthy discs. However, particularly with respect to low-grade slip or assessing changes between time-points, radiographic measurements of slip are imperfect [1,2,3,4]. Reliable and standardized diagnosis requires robust reference data to differentiate between normal and abnormal cases, and an understanding of relevant co-factors. The same challenges apply to disc space narrowing (DSN).

Currently, objective criteria for reproducibly classifying intervertebral disc metrics are poorly developed. The challenge is to validate widely applicable criteria for classifying “normal” versus “abnormal” such that these metrics can be consistently used, along with other clinical details, in support of clinical decision making.

The rapid advancement of artificial intelligence/machine learning technology in spine care enables near instant, unbiased quantitative measurements [5,6,7,8]. To utilize this technology effectively, validated guidelines are needed to standardize testing, minimize errors, and interpret the relationship between automated measurements and clinical outcomes.

The 2nd National Health and Nutrition Examination Survey (NHANES-II), conducted between 1976 and 1980, collected over 7000 lateral lumbar spine X-rays as part of a nationwide probability sample [9]. These X-rays, along with demographic, anthropometric, health, and medical history data, were collected to document the health status of the United States. NHANES-II data may be valuable for establishing normative reference data in clinical practice and research studies.

With the goal of establishing reference data for the diagnosis of abnormal SPO and disc space narrowing, the aims of this study were as follows:Determine if objective disc metrics can be developed that account for (1) the variability due to patient positioning, (2) the normal variability between patients, and (3) the variability between levels, using the database of over 7000 lumbar spine radiographs from the NHANES-II study, supplemented with a collection of flexion-extension radiographs to better understand the importance of disc angle and load bearing.Develop reference data for diagnosis of abnormal disc metrics based on the NHANES-II radiographs by excluding degenerated discs and outliers in the data distributions.Document the errors that can occur in disc metrics due to variability in radiographic projection using precisely calculated landmarks obtained from variable, digitally reconstructed radiographic projections.

## 2. Materials and Methods

NHANES-II images and participant data were obtained using a public access protocol (https://www.cdc.gov/nchs/nhanes/index.htm accessed on 12 September 2022). The Spine CAMP^TM^ software (Medical Metrics, Inc., Houston, TX, USA), a previously validated and FDA-cleared image analysis platform, was used to automatically obtain four landmarks for each vertebral body from L1 to S1. Details of landmark placement can be found in Appendix A, as proper placement is crucial for accurate SPO and disc measurements. Vertebral morphology based on these landmarks has been reported previously [10]. While specific software was used to obtain landmarks, the focus of this study was not on the software itself, as the landmarks are intended to be consistent with those used in prior research studies [11,12,13,14,15]. Multiple machine-learning approaches have been used to produce vertebral landmarks in prior studies [16,17,18,19,20,21], and it is expected that other automated methods can reproduce the landmark placement obtained with the Spine CAMP^TM^ software for NHANES-II radiographs.

With the goal of reserving the use of the term “spondylolisthesis” for abnormal sagittal plane positioning between vertebrae, the term “sagittal plane offset” (SPO) was used generically in the following for all measurements of anterior-posterior displacement of one vertebra with respect to the adjacent vertebra. It was assumed that a range of SPO would be “normal” and SPO outside of that range can be labeled spondylolisthesis (either anterior or posterior).

In addition, a neural network was trained to identify osteophytes and sclerosis. This was used to help exclude degenerated discs from the definition of normal disc metrics. The neural network, the training/validation of the network, and the results are described in Appendix A.

The following nine metrics were calculated from the landmarks for each intervertebral level from L1-L2 to L5-S1, using one of two methods, which were as follows:Method 1○Anterior disc height (ADH);○Posterior disc height (PDH);○Disc angle (DA);○Anterior SPO (ASPO);○Posterior SPO (PSPO).Method 2○Ventral disc height (VDH);○Dorsal disc height (DDH);○Mid-plane angle (MPA);○Centroid SPO (CSPO).

The average disc height was also calculated as the average of ADH and PDH. A disc area metric was also calculated (in units of endplate width squared) from the four landmarks using the shoelace formula.

Method 1 metrics are illustrated in Figure 1. Landmarks obtained using method 1 are intended to represent the mid-sagittal plane of the vertebral body. Method 2 metrics were all initially described by Frobin et al. [13,22] with the goal of minimizing the influence of radiographic distortion on measurements, and they are illustrated in Figure 2 and Figure 3. Since substantial variability can occur in radiographic magnification [23,24,25], and since there was no scaling device in the NHANES-II X-rays, all SPO and disc height metrics were reported in units of % anterior-posterior endplate width using the superior endplate of the inferior vertebra. A positive ASPO reveals that the anterior inferior corner of the superior vertebra is anterior to the anterior superior corner of the inferior vertebra. A positive PSPO reveals that the posterior inferior corner of the superior vertebra is posterior to the posterior superior corner of the inferior vertebra. A negative DA or MPA reveals that the ADH (or VDH) is smaller than the PDH (or DDH).

Initially, means, standard deviations, skewness, and kurtosis were generated including all discs to assess data distributions. Most metrics had significant skewness and kurtosis indicating non-normal data distributions. With the assumption that normal sagittal plane offset (SPO) and other disc metrics should not include levels with definite osteophytes and/or sclerosis, those levels were identified using data from the neural network described in Appendix A. Based on work to develop a spondylolisthesis index (Appendix A), the ratio of the width of the inferior endplate of the superior vertebrae over the width of the superior endplate of the inferior vertebra was calculated and used with the intention of excluding levels where one vertebra was significantly wider than the adjacent vertebra. Levels were also excluded if the height over the width of the superior vertebra was outside the 95% confidence interval. This was to exclude fractured vertebrae or other abnormal vertebral body morphometry. Means, standard deviations, skewness, and kurtosis then regenerated for each metric, excluding all discs with definitive osteophytes or sclerosis. The means and standard deviations were then used to determine the upper and lower limits of the 95% confidence interval for each metric. Final descriptive statistics were generated including only levels with no definitive osteophytes or sclerosis and where all of the following metrics were within the 95% confidence intervals: disc angle, ASPO, PSPO, average disc height, disc area, endplate width ratio, height over width ratio of the superior vertebra. These data are intended to be used as reference data to help identify abnormal metrics. Pearson’s correlation coefficients were used to help identify the strength and direction of associations between metrics, with >0.7 defined as a strong correlation, and very strong correlation defined as a coefficient > 0.9 [26].

Since Z-scores simplify the interpretation of metrics for an individual, the final descriptive statistics were used to generate “Z” scores for each metric:(1)$$Z\;\mathrm{score}=\frac{disc\;metric-average\;for\;normal\;discs}{std\;dev\;for\;normal\;discs}$$

Based on the potential advantages of obtaining descriptive statistics that resist outliers, robust statistics were generated using the Hodges–Lehman estimate of location to determine the central tendency, and the Qn statistic was used to determine the spread (robstat package http://github.com/benjann/robstat/ accessed on 10 September 2023) [27,28]. The robust statistics were used to generate a normalized score analogous to a Z-score:(2)$$\mathrm{Normalized}\;\mathrm{score}=\frac{disc\;metric-HodgesLehman\;estimate\;of\;location}{Qn\;statistic}$$

Multivariate analysis of variance was conducted to explore relationships between disc metrics and age, sex, BMI, and other variables. All statistical analyses were performed using Stata version 15 (College Station, TX, USA).

Since all NHANES-II radiographs were acquired with the participant in a side-lying position, there was limited variation in flexion angles compared to clinical practice. To better understand the impact of disc angles and disc heights on SPO, flexion-extension radiographs from a prior study [29] were analyzed to obtain a wider range of disc angles. Regression analysis with post-estimation calculations yielded coefficients for predicting normal SPO from flexion–extension radiographs, based on the most predictive variables and excluding levels identified as outliers. This prediction equation led to the development of a Spondylolisthesis Index (SpondyIndex), intended to be useful for detecting abnormal SPO in both NHANES-II and flexion–extension X-rays. Details of the SpondyIndex development are provided in Appendix A.

It is recognized that variability in radiographic projection can impact spine measurements [22,30]. To assess measurement error in the nine SPO and disc metrics, simulated X-rays were generated with precise knowledge of the true landmark locations. Details of this experiment are described in Appendix A.

The NHANES lumbar X-rays were obtained in a side-lying position whereas in clinical practice, many X-rays are obtained with the patient standing. Differences in spinal loading between side-lying and standing positions are not well understood. To determine the disparity between reference disc metrics from side-lying versus standing neutral X-rays, 2866 pre-operative neutral-lateral X-rays obtained in the standing position were also analyzed. The landmark coordinates were obtained from X-rays using the same fully automated methods employed for NHANES X-rays: vertebral landmark identification, neural network-based disc degeneration grading, exclusion of degenerated and statistical outlier discs, and descriptive statistics generation for “normal” discs measured from standing X-rays.

In the NHANES II study, subjects were asked if they had experienced back pain on most days for at least two weeks. Logistic regression was used to examine the effects of disc metrics on reported back pain.

## 3. Results

### 3.1. Summary of Data Analyzed

Sagittal Plane Offset (SPO) and disc metrics were obtained from lumbar X-rays for 36,051 levels in 7414 of the 7422 NHANES-II. One NHANES-II file was corrupt, and analysis of seven of the NHANES X-rays was not feasible due to the inability to identify landmarks for at least two contiguous vertebrae. Generating landmarks for all X-rays took approximately 9 h on a dual CPU and 4 GPU (Nvidia T4) production server, after which disc and SPO metrics were calculated. For 1041 levels (2.8% of total possible levels), disc metrics were not produced due to issues such as incomplete inclusion of vertebrae in the field of view or identified issues by the image analysis software (overlying radiographic labels, artifacts from clothing, etc.). Age and body size data are summarized in Table 1, with older ages being disproportionately represented, as shown in Figure 4.

### 3.2. Establishing Normal Disc Metrics

The normalized scores using the Hodges–Lehman estimate of location and the Qn scale statistic were nearly identical to the standard Z scores using means and standard deviations (SD). Since the standard Z-score would be more familiar to most potential users of the metrics, standard descriptive statistics were used instead of the robust statistics. Appendix A, provide descriptive statistics for the NHANES-II lumbar SPO and disc metrics, before and after trimming the data. Note that trimming (1) eliminates approximately half of the data, (2) does not dramatically change the mean or median, (3) substantially lowers the standard deviation and coefficient of variation, and (4) brings skewness close to 0 and kurtosis closer to 3 (characteristics of a normal distribution).

### 3.3. Correlations between Disc and SPO Metrics

The disc and sagittal plane offset (SPO) metrics for method 1 and method 2 showed significant (*p* < 0.05) correlations with each other. Pearson correlation coefficients for the L3L4 level are presented in Table 2. The R^2^ values were particularly high (>0.9) between anterior disc height and ventral disc height, posterior disc height and dorsal disc height, as well as average disc height and disc area. Additionally, the method 1 disc angle and method 2 mid-plane disc angle demonstrated a strong correlation.

It is worth noting that ASPO (anterior SPO) and CSPO (central SPO) displayed a positive correlation, while PSPO (posterior SPO) and CSPO showed a negative correlation. This outcome arises from PSPO quantifying the translation of the posterior–inferior corner of the superior vertebra into the spinal canal.

### 3.4. Analysis of Sources of Variance in SPO Metrics

Based on a multivariate analysis of variance (ANOVA), after excluding levels with osteophytes/sclerosis or abnormal metrics, ASPO and PSPO (method 1) were significantly dependent on the disc area, endplate width ratio, anterior and posterior disc heights, disc angle, level, age, BMI, and nation of origin (R^2^ = 0.95, *p* < 0.02 for all variables). Sex and race were not significant for ASPO (*p* > 0.21 for both). Dropping demographic variables lowered the R^2^ minimally (0.946 to 0.945). With demographics excluded, based on the F-values, disc area, followed by endplate width ratio, disc heights, disc angle, and level, had the strongest associations with ASPO. Dropping the disc angle lowered R^2^ only slightly to 0.941, supporting that use of anterior and posterior disc heights can account for the variability in disc angles. The R^2^ for PSPO using the same variables in ANOVA was 0.9. Sex, race, and nation of origin had a significant but minimal association with PSPO. Dropping demographic variables changed the R^2^ for PSPO minimally.

CSPO (Method 2) was significantly dependent on level, ventral and dorsal disc heights, endplate width ratio, age, BMI, sex, and midplane angle (*p* < 0.0001 for all variables). CSPO was not significantly dependent on race or nation of origin (*p* > 0.49). Intervertebral level was over an order of magnitude more important than any other variable. The F-value for midplane angle, BMI, and sex were small. Including only level, R^2^ = 0.72, and this was only slightly lower than when all variables were included (R^2^ = 0.74). Thus, with CSPO, over 70% of the variability in CSPO can be explained by level alone. 

The standardized expression of average disc height, in units of standard deviations from average normal, was linearly related to age (*p* < 0.0001), but the R^2^ was low (0.032). An exponential curve fit (*p* < 0.0001 for both coefficients) had a higher R^2^ (0.12):(3)$$zAvgDH=b0\times e^{\left(b1*age\right)},$$
where *b*0 = −0.0037, *b*1 = 0.078.

### 3.5. Applicability of NHANES-II Reference Data to Flexion-Extension X-rays

The NHANES-II IVM reference data, obtained from participants imaged in a side-lying position, were applied to 161 flexion–extension radiographs of asymptomatic volunteers (imaged in an upright position, as described in Appendix A). The application of the standardized average disc height metric (zAvgDH), developed from the post-trimmed NHANES-II reference data, to the asymptomatic flexion–extension X-rays, revealed 32 levels (out of a total of 800 levels) with zAvgDH < −3 (based on the average between the flexion and extension X-rays), indicating objective evidence of substantial disc narrowing. Among these 32 levels, the neural network (described in Appendix A) detected osteophytes/sclerosis in 28 (88%) of the levels. It is not surprising to find that disc height narrowing and osteophytes/sclerosis tend to coexist, supporting the use of standardized average disc height from NHANES-II data to identify abnormal disc height loss in flexion–extension X-rays.

However, the NHANES-II reference data for SPO did not yield satisfactory results with the flexion–extension X-rays: 53% of the flexion and extension radiographs had standardized (using NHANES-II) ASPO or PSPO metrics >2 SD from the average normal. Given that 53% of asymptomatic levels cannot have abnormal SPO, it is evident that applying the NHANES-II normative SPO data to flexion and extension X-rays is inappropriate. It is unsurprising as a small range of disc angles, as observed in the NHANES study, cannot be extrapolated to the large range of disc angles found in flexion–extension studies. This result also documents the dependence of SPO on disc angle.

The initial hope was to develop standardized SPO metrics to identify abnormal SPO across a wide range of spine positions, acknowledging that abnormal SPO may only appear with certain spinal positions. In a normal spine, the ideal, standardized, SPO metric would be similar when applied to either flexion or extension X-rays. However, substantial differences were observed in the predicted (using NHANES-II data) SPO between the flexion and extension X-rays for asymptomatic volunteers with radiographically normal spines. This confirmed that the standardized ASPO, PSPO, and CSPO metrics, developed from NHANES-II X-rays, only account for the relatively low variability that occurred in a collection of X-rays obtained in the lateral decubitus position. As a result, the standardized NHANES-II SPO data are not applicable to X-rays taken in a flexed or extended position unless the thresholds used to classify a level as normal or abnormal are set at a high level.

### 3.6. Applicability of a SpondyIndex Based on Flexion-Extension X-rays to NHANES-II X-rays

With the aim of developing a method that accurately predicts normal sagittal plane offset (SPO) by considering disc angles, disc heights and other metrics, Appendix A introduces a “Spondylolisthesis Index” (SpondyIndex) derived from flexion–extension X-rays of asymptomatic volunteers. The SpondyIndex can estimate normal SPO while accounting for variations in disc area, endplate width ratio (EPWR), disc heights, and the spinal level. This comprehensive approach makes it applicable to both NHANES-II X-rays and flexion–extension X-rays.

The SpondyIndex was reported in units of deviation from average normal, using radiographically normal levels from asymptomatic volunteers as the reference. A SpondyIndex value > 2 indicates a measurement above the 95% confidence intervals for normal SPO, while a value > 3 represents a substantially higher deviation from normal. Applied to the NHANES-II data, 3.2% of levels had an anterior SpondyIndex > 2, and 1.6% had an anterior SpondyIndex > 3. In addition, 2.9% of levels showed a posterior SpondyIndex > 2, and 0.59% had a posterior SpondyIndex > 3. A posterior SpondyIndex > +2 suggests the presence of retrolisthesis. The SpondyIndex thus offers a potentially valuable tool to objectively identify abnormal SPO and specific conditions like retrolisthesis, enhancing the understanding of spine pathology and improving diagnostic accuracy.

### 3.7. Prevalence of Abnormalities in the NHANES-II Lumbar Spine Radiographs

Using the post-trimming mean and standard deviation data in Appendix A, each measurement in the NHANES-II study was classified as “normal” or “abnormal”. The prevalence of abnormalities in the NHANES-II data is also provided in Appendix A. Prevalence was calculated using the standardized (“Z”) version of the metrics to simplify interpretation of the data. The Z score informs how many standard deviations a measurement was above or below the mean for normal discs. This was carried out on a level-specific basis. Both the prevalence of Z scores > 2 and >3 are provided, since a Z score of 2 is just outside the 95% confidence interval whereas a Z score of 3 is more definitively abnormal. There was generally a substantially lower percentage of levels where the Z score was over 3 versus over 2 standard deviations from average normal.

### 3.8. Associations with Back Pain

In the NHANES II study, participants were asked if they had experienced back pain on most days for at least two weeks. Among the participants, 18.1% reported back pain. Investigating associations between the response to this question and the disc and sagittal lane offset (SPO) metrics poses a challenge as it is unknown which level(s), if any, in the spine might be associated with such back pain. Despite this, we calculated the maximum anterior spondylolisthesis index (AntSI) and the maximum posterior spondylolisthesis index (PstSI) across all levels for each spine and used logistic regression to examine their association with reported back pain. The analysis revealed a significant association between maximum PstSI and the presence of back pain (*p* < 0.0001). However, the odds ratio was 1.14, and the R^2^ value was very small (0.0022). This indicates that while an association between SPO and back pain can be detected using NHANES-II data, very little variability in back pain can be explained by SPO.

Likewise, we computed the minimum standardized average disc height for each participant and observed a significant association with reported back pain (*p* < 0.0001). The odds ratio (0.80) suggests a slightly stronger association between back pain and disc height loss compared to SPO. Nevertheless, the R^2^ was also very small (0.014), indicating that a very small amount of variability in back pain can be explained by disc height loss. In summary, although some weak associations between SPO, disc height loss, and back pain can be identified in the NHANES-II data, the overall relationships are minimal. It is possible that abnormal SPO or substantial disc height loss can contribute to symptoms though the NHANES data are not ideal for testing that hypothesis.

### 3.9. Descriptive Statistics Disc Metrics Measured from Standing Neutral-Lateral X-rays

Supplementary File S6 provides the results of reference data generated from the analysis of neutral-lateral X-rays pooled from 12 independent studies. Appendix A was organized the same as Appendix A so that comparisons can be more easily made of the differences in reference data established from the NHANES X-rays (participants side-lying) versus from neutral-lateral standing X-rays. Comparison of trimmed data (numbers in brackets) in Appendix A to analogous tables in Appendix A reveal some similarities, and some differences that may be explained by the differences in spinal loading between side-lying and standing.

Comparing data in Appendix A, the following trends were observed: standing X-rays generally exhibited higher anterior disc heights (after excluding degeneration and outliers) than side-lying X-rays, except at L4-L4 and L5-S1. This could be explained by the fact that NHANES patients were somewhat flexed in the side-lying position. On the other hand, posterior disc heights were lower in standing X-rays, likely influenced by compressive loads on the spine and the semi-flexed side-lying position. However, average disc heights remained consistent within 1% of endplate width at all levels, except for L5-S1, where the average was 2% lower in standing X-rays than side-lying. For assessing loss of disc height, average disc height serves as valuable reference data, with little difference whether side-lying or standing X-rays are used. The NHANES-II reference data hold the advantage of representing a broader population and having accompanying data on sex, age, BMI, and more.

Additionally, comparing the prevalence data in Appendix A to analogous tables in Appendix A yields interesting insights. These data include all participants and levels, not just those without degeneration or outliers. Not surprisingly, more abnormalities are documented in Appendix A (symptomatic patients enrolled in clinical trials) compared to Appendix A (a statistical sampling of the US population). This finding supports the potential of the metrics to detect abnormalities, particularly in scenarios where a higher proportion of abnormalities would be expected, such as clinical trial patients with specific symptoms. The comparison of reference data between side-lying and standing X-rays, as well as the examination of prevalence data, provides valuable insights into the potential applications and limitations of the metrics in different contexts.

## 4. Discussion

Data representing a large population are needed to establish benchmarks for normal disc metrics, enabling the identification and diagnosis of abnormal disc metrics. The NHANES database of X-rays presents a potential opportunity to establish a standardized definition of normal sagittal plane offset (SPO) and other disc metrics applicable across research studies and clinical practice. Ideally, this definition should be valid for both neutral-lateral and flexion–extension X-rays, reducing dependence on patient positioning. To achieve this goal, the NHANES-II data, obtained with participants in the lateral-decubitus position, were analyzed to define normal SPO and other disc metrics. To improve generalizability, a smaller database of lumbar flexion–extension X-rays was also examined to identify variables that best predict normal SPO. To further understand dependence on patient position, standing neutral-lateral X-rays were analyzed.

The analysis of NHANES-II data revealed strong correlations between certain SPO and disc metrics, while others showed weaker correlations, indicating unique aspects of disc height, angle, and SPO detection. A prediction equation developed using flexion–extension X-rays of asymptomatic participants performed well with both NHANES-II and flexion–extension studies, enabling better predictions of normal SPO while considering disc heights and angles. Further research is needed to determine how to effectively use reference data for SPO and other disc metrics in research and clinical practice.

Among multiple SPO and disc metrics investigated, CSPO emerged as a good metric for overall SPO assessment, while ASPO and PSPO might be more effective in assessing local effects of SPO. The difference in research and clinical efficacy between the method 1 and method 2 SPO measurements requires further investigation.

A primary goal was to establish normal sagittal plane SPO to identify and quantify abnormal SPO (spondylolisthesis). Due to limited resources, manual clinician assessment of spondylolisthesis in all NHANES-II X-rays was not possible, preventing sensitivity/specificity analysis of automated measurements against a standard clinical assessment. Even if manual assessments of disc height, angles and SPO had been obtained, error due to observer variability would be expected [2,31,32,33].

While expert consensus favors standing neutral-lateral radiographs for spondylolisthesis diagnosis, it can also be assessed from side-lying or flexion and extension X-rays [34,35,36]. Limited evidence exists on how these positions affect spondylolisthesis assessment. Considering the Pearson correlation coefficients in Table 2, all three SPO metrics were significantly correlated with disc angle, supporting that positioning effects SPO. However, the NHANES-II data, along with the analysis reported in Appendix A, supports that the positioning effects can be accounted for. The SpondyIndex emerged from this analysis, serving as a predictive tool for normal SPO while considering disc area, EPWR, disc heights, and level.

The experiment in Appendix A documented errors in disc height, angle, and SPO measurements due to variability in radiographic projections. For instance, the median error in measuring the standardized average disc height from a radiograph was 0.13 standard deviation from the average normal disc height. This variability was expected due to radiographic projection. However, in cases of highly out-of-plane imaging of a disc space, errors can be as high as 1 standard deviation solely due to the poorly imaged level.

To address this issue with lateral spine radiographs, potential solutions include optimizing imaging protocols for each level and possibly using neural networks trained to correct for out-of-plane imaging. Until such solutions are developed, caution should be exercised when interpreting disc metrics for severely out-of-plane imaging of lumbar disc spaces.

When comparing data between studies, it is essential to consider that multiple approaches to imaging and measuring spondylolisthesis may have been used [37,38]. Currently, there is no widely validated consensus on the best method for measuring and interpreting spondylolisthesis. Specific criteria for interpreting spondylolisthesis measurements must be tailored to the method used to avoid misclassifying it as normal or abnormal [38]. The reliability of the measurements depends on the method employed, and measurement error has been documented in previous studies [2,3,38,39,40,41,42,43]. Reproducibility, as indicated by interclass correlation coefficients, is one aspect, but accuracy is equally crucial.

Optimizing the diagnosis of spondylolisthesis also depends on understanding the variables that influence the interpretation of measurements. In clinical studies, reported treatments for symptomatic spondylolisthesis may lack details about how the assessment was made, except for a threshold amount of SPO (e.g., Austevoll et al. [44]). Using a threshold level of SPO, when SPO is measured in millimeters, requires correction for image magnification, which can vary significantly between X-rays [23]. To avoid errors when radiographic magnification is unknown, it may be preferable to measure SPO as a percentage of endplate width and use appropriate reference data.

In addition to measuring the magnitude of SPO, other aspects of spondylolisthesis may be clinically relevant [45,46]. The degenerative form is associated with arthritis and facet joint hypermobility [47,48,49]. Hypoplasia of L5 can also mimic spondylolisthesis [4,50,51]. The spondylolisthesis index in Appendix A utilizes the endplate width ratio to mitigate endplate width discrepancies. NHANES-II vertebral morphology data can help identify abnormalities in morphology [10]. Considering factors like pelvic incidence may also be important when interpreting SPO [41,52]. An alternative subjective classification system by Marchetti-Bartolozzi exists [53]. The relative importance of SPO magnitude versus other classification methods is yet to be determined. Artificial intelligence, with sufficient training material, could be used to objectively apply different subjective spondylolisthesis classification systems, potentially enhancing quantitative metrics.

Previous studies have assessed the prevalence of abnormal SPO and can help with interpreting the prevalence data in Appendix A. Kalichman et al. found a 21% prevalence in the Framingham heart study, with a correlation to age [54]. However, their sample and assessment methods differ from NHANES-II, as they used CT exams. Occupational exposure may also be associated with abnormal SPO [55], which could be tested using the SpondyIndex. Notably, Cehn et al. found disc height and posterior-to-anterior vertebral body height ratio to be predictors of spondylolisthesis [56], suggesting potential insights if repeated with objective and standardized SPO metrics.

The disc height data from NHANES-II lumbar X-rays can aid in assessing disc height loss. Although the NHANES-II data represent disc heights in the side-lying position, applying this reference data to upright flexion–extension X-rays did not yield unexpected results. The average disc height reference data were very similar when developed using side-lying versus standing X-rays. Although diurnal changes in disc heights have been documented [57,58], the NHANES-II X-rays were presumably collected without specific control over the daily loading history up to the time of imaging, so it is likely the NHANES-II data represent a wide range of diurnal variation, and thereby account for this variation in the standard deviations. In the analysis of the NHANES-II radiographs, there was an exponential relation between age and the standardized average disc height metric (*p* < 0.0001, R^2^ = 0.12) such that by age 74, the standardized disc height would be expected to be −1.4. It is worth noting that other studies have reported varying associations of disc height with age, with the strength of the association being crucial to consider [59,60,61,62,63,64]. Compared to studies showing a difference in disc height between males and females when disc height was measured in millimeters [63,64], expressing disc height in percent endplate width may have minimized differences between sexes. By reevaluating previous studies using standardized and automated metrics, our understanding of age effects on intervertebral discs may be improved.

Publications and spine fusion coverage policies often suggest that spondylolisthesis and instability are interchangeable, but several studies have proven this to be untrue [65,66,67,68,69]. The NHANES-II data cannot address this issue due to the lack of validated diagnostic metrics for instability [70,71,72] and the use of only static X-rays. The clinical importance and optimal treatment for spondylolisthesis remain incompletely understood [73,74,75,76], and the assessment of whether the spondylolisthesis is static or dynamic is likely crucial [77,78].

Limitations of this study include the following:(1)Poor representation of some races, nations of origin, and sex in some age groups. In particular, females are under-represented. By design of the NHANES-II study, lumbar X-rays were not intended to be obtained for pregnant women or women under 50 [79]. An additional limitation relates to the data on race and nation of origin recorded in the NHANES-II data. Of the NHANES-II participants, 86.9% were “white”, 11.2% “black”, and the rest “other”. A more uniform representation of races would likely be needed to fully understand the importance of race. The same is true for the “nation of origin” data in NHANES-II. The ages of the participants were also biased toward older ages.(2)NHANES-II lumbar spine X-rays were obtained with participants lying on their sides. Disc metrics in Appendix A may not be directly and precisely applicable to X-rays taken in other positions. Using all the NHANES data, the L1-S1 angle was 51.3 (SD 12.8). Table 3 provides data from three prior studies reporting L1-S1 angles measured from X-rays obtained with participants standing [80,81,82]. The lordosis data from the NHANES-II study was very similar to lordosis measured from standing X-rays in three other studies. Figure 5 shows mid-plane angles from the NHANES-II study, excluding levels where any disc or SPO metric was abnormal, compared to mid-plane angles reported by Frobin et al. from X-rays obtained with participants standing [13]. There are some differences, but also some similarities. These comparative data can help in deciding whether to use NHANES-II reference data with standing or other X-rays. With respect to external validity of the disc height measurements, comparable reference data were hard to find. There are multiple publications reporting lumbar intervertebral disc height reference data in units of millimeters, frequently from MRI or CT exams [60,63,83,84]. Since a scaling device was not included in the NHANES-II X-rays, these prior publications cannot be used as comparative data. Data for a large collection of radiographs obtained in other positions and analyzed using the same methods would be required to understand the applicability of the NHANES-II data to other protocols for obtaining lateral lumbar spine X-rays. It was assumed that the average disc height data would be the most universally applicable of the NHANES-II disc height reference data, since average disc height should minimize differences due to disc angle changes from variability in patient positioning.

(3)The accuracy study using simulated X-rays (Appendix A) revealed potentially large errors in disc and SPO metrics when the radiographic projection was very poor. It would be valuable to establish a neural network or other method to either make a correction in the metrics when large out-of-plane imaging is encountered, or to abstain from reporting data when vertebrae are poorly imaged. Nevertheless, a certain level of error must be expected in SPO and other metrics. This may be a particularly significant issue with large amounts of frontal plane spinal curvature. This study does not provide clear guidance on this issue and additional data are needed. Caution should be used when interpreting disc metrics and SPO from lateral X-rays where the vertebrae are poorly imaged.(4)Trimming the data such that only truly normal discs were used to define “normal” disc and SPO metrics was a challenge. Abnormal disc heights, angles, and SPO can be expected in the NHANES-II study (since there was no attempt to exclude spine abnormalities). It was assumed that normal disc heights, disc angles, and SPO would have a Gaussian distribution. The NHANES data were also analyzed with what are referred to as “robust” statistics [27,28]. Standardized scores were generated from the robust statistics, but these were nearly the same as the standard Z scores generated using means and standard deviations. Since the standard Z scores are easier to explain in diagnostic tests, robust statistics were not used. In addition, in a normal disc, the resting position of the vertebrae, when the radiograph was obtained, would be expected to be in the neutral zone. Within the neutral zone, little force is required to produce sagittal plane movements [85,86]. It can be hypothesized that the amount of SPO may change slightly, but remain within the neutral zone, every time a person assumes a “neutral” position. That would be expected to contribute to a normal distribution of SPO within a large population. There is currently no gold standard method that is validated for classifying discs as normal versus abnormal. MRI exams with appropriate imaging sequences would have provided a more robust assessment of disc health, but they were not available. In the current study, an attempt was made to trim data so as to achieve a Gaussian distribution. This approach has been used by other authors [87,88,89]. A better strategy may be possible for assuring that only truly normal discs are used to define normative reference data. For example, a formal optimization scheme might be used to obtain the best possible Gaussian distribution, though justification for such an optimization is not well developed.(5)Although Appendix A provides data to calculate disc and SPO metrics as number of SD from average normal, a clinically meaningful threshold must be validated for use with these reference data that can classify a metric as normal vs abnormal. Even though a metric that is two SD from average would be technically outside of the 95% confidence interval used to define “normal”, that may not be clinically significant. Until well-designed clinical trials are completed, the threshold level for the standardized score that is clinically efficacious will not be known.

## 5. Conclusions

The NHANES-II radiographs were used to develop reference data to aid in interpreting measurements of sagittal plane offset (SPO), disc heights, and disc angles as normal versus abnormal. These metrics are dependent on disc level, so they should always be interpreted with level-specific reference data. Within the NHANES-II data, normal SPO could be predicted from disc heights, endplate width ratios, and disc area. However, application of the SPO predictions from the NHANES data to flexion–extension studies revealed limitations, likely related to the low variability in disc angles in the NHANES-II radiographs. However, normal SPO predictions developed using flexion–extension radiographs for asymptomatic volunteers work well for identifying abnormal SPO in the NHANES study. A “SpondyIndex” was developed that is simple to interpret: an anterior SpondyIndex > 2 objectively identifies anterior listhesis, and a posterior SpondyIndex > 2 identifies retrolisthesis. An analysis was completed that documents a need for caution in use of disc metrics when the X-ray is substantially out-of-plane. Although the reference data developed from the NHANES-II lumbar X-rays may help to identify and quantify abnormal discs, they would need to be tested for efficacy in appropriate clinical trials. Multiple metrics were assessed using the NHANES-II data, yet it will take further research to determine which, if any, are clinically efficacious. The greatest challenge is to validate diagnosis and treatment algorithms that can effectively make use of disc and SPO metrics to improve outcomes for patients with spinal disorders.

## Figures and Tables

**Figure 1 bioengineering-11-00360-f001:**
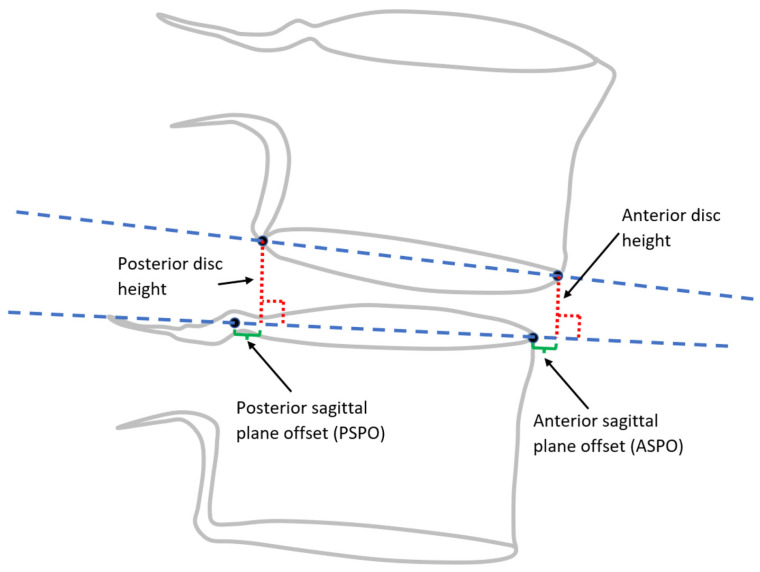
Disc height and SPO measurements using method 1. The blue dashed lines representing the mid-sagittal planes of the endplates. The red dotted lines are perpendicular to the blue dashed line on the superior endplate of the inferior vertebra and represent the anterior and posterior disc heights.

**Figure 2 bioengineering-11-00360-f002:**
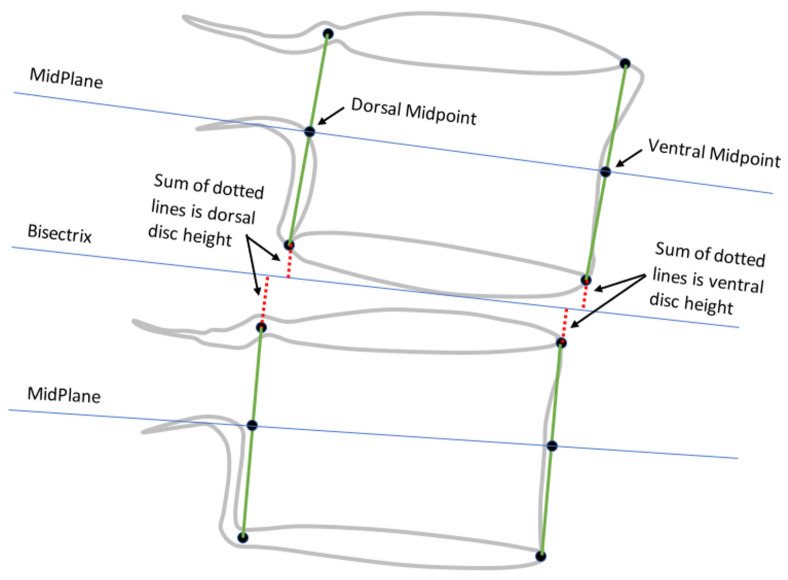
Disc height measurements using method 2 (adapted from Frobin et al. [13]) Blue lines show vertebral midplanes and disc bisectrix. Red dotted lines show disc height measurements.

**Figure 3 bioengineering-11-00360-f003:**
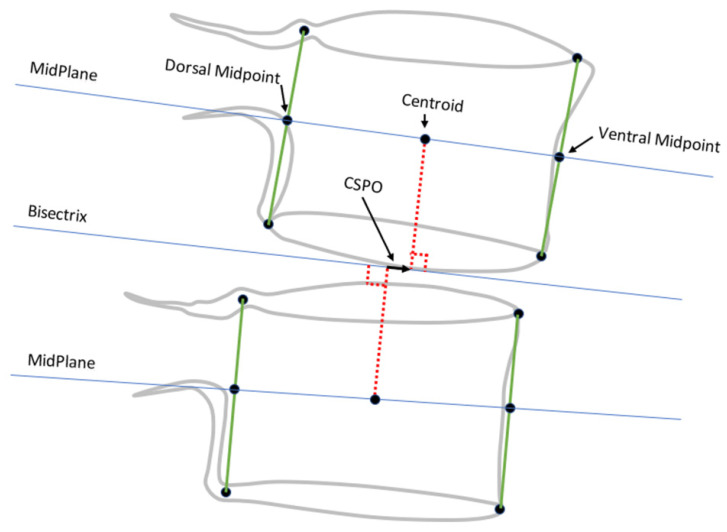
Method used to measure centroid sagittal plane offset (CSPO), adapted from Frobin et al. [22]. The red dotted lines connect the centroids along the blue midplane lines to the disc bisectrix and are perpendicular to the bisectrix. CSPO therefore measures the offset between the vertebral centroids along the disc bisectrix.

**Figure 4 bioengineering-11-00360-f004:**
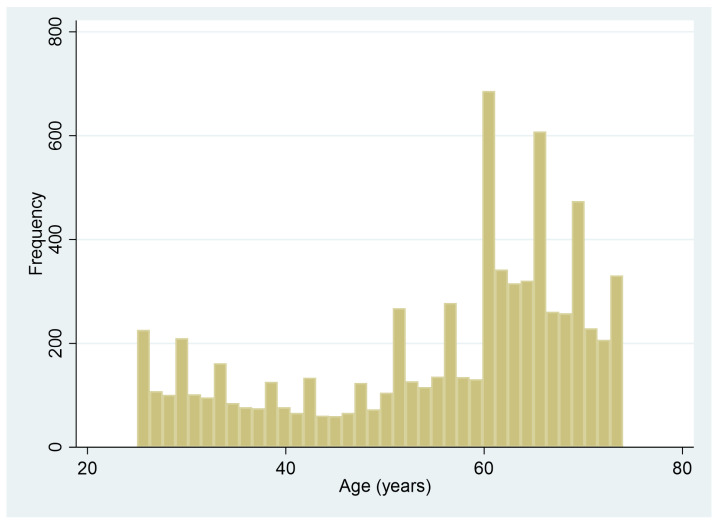
Distribution of ages in the NHANES-II study.

**Figure 5 bioengineering-11-00360-f005:**
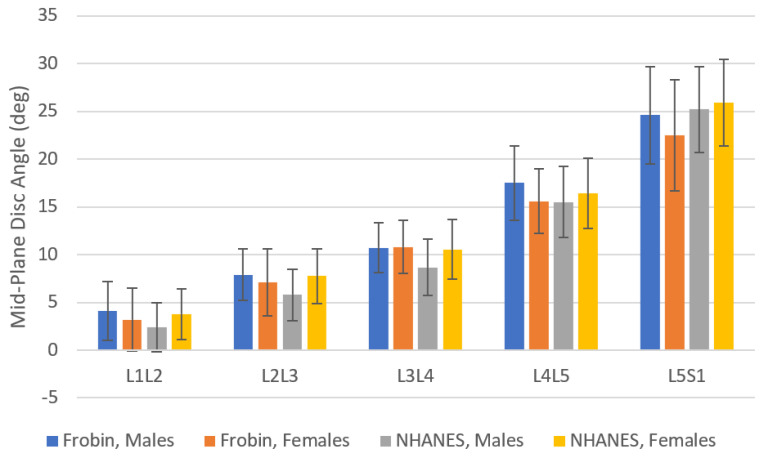
A comparison of mid-plane angles from the NHANES-II study (excluding all levels where any metric was abnormal) to data from Table 3 of Frobin et al. [13]. X-rays in the NHANES-II study were obtained with participants lying on their sides in a semi-flexed position, while Frobin et al. obtained X-rays with patients standing. The error bars show standard deviations. The sample size in Frobin et al. was 61 to 239 depending on sex and level, while the sample size in the NHANES-II data was 1076 to 2735 depending on sex and level.

**Table 1 bioengineering-11-00360-t001:** Sample size, average age, and standard deviation (SD), and average Body Mass Index (BMI) in units of kg/m^2^ for the NHANES-II participants that were analyzed. Two participants did not have demographic data. The SD is in brackets.

Sex	No.	Age	BMI
Male	4582	50.9 [15.3]	25.5 [4.0]
Female	2830	63.3 [6.4]	26.4 [5.5]

**Table 2 bioengineering-11-00360-t002:** Pearson correlation coefficients between each pair of variables, for the L3L4 level, including data for all participants. Coefficients ≥ 0.8 are bolded. Similar results were found at all other levels. See Methods Section for definition of acronyms.

	ADH	PDH	Avg DH	DA	ASPO	PSPO	VDH	DDH	MPA	CSPO	Disc Area
ADH	1										
PDH	0.48	1									
AvgDH	**0.9**	**0.81**	1								
DA	0.7	−0.29	0.32	1							
ASPO	−0.33	0.19	−0.12	−0.53	1						
PSPO	0.29	−0.07	0.16	0.36	−0.78	1					
VDH	**0.99**	0.45	**0.88**	0.72	−0.4	0.29	1				
DDH	0.5	**0.99**	**0.82**	−0.26	0.12	−0.05	0.48	1			
MPA	0.52	−0.31	0.19	**0.82**	−0.37	0.2	0.54	−0.29	1		
CSPO	−0.17	0.11	−0.06	−0.28	**0.82**	**−0.86**	−0.21	0.06	−0.19	1	
Disc Area	**0.91**	0.79	1	0.34	−0.15	0.22	**0.88**	0.8	0.21	−0.11	1

ADH, Anterior Disc Height; PDH, Posterior Disc Height; Avg DH, Average Disc Height Metric; DA, Disc Angle; ASPO, Anterior Sagittal Plane Offset; PSPO, Posterior Sagittal Plane Offset; VDH, Ventral Disc Height; DDH, Dorsal Disc Height; MPA, Mid-Plane Angle; CSPO, Centroid Sagittal Plane Offset.

**Table 3 bioengineering-11-00360-t003:** L1-S1 angles (degrees) from three prior studies [80,81,82] measured from standing lateral X-rays compared to L1-S1 angles in the NHANES-II study.

	N	Mean Age	Mean L1-S1	SD L1-S1
Korovessis [80]	45	63	48	12.5
Lord [81]	109	47	49	15
Been [82]	100	38	51	
NHANES-II	6397	55	51	13

## Data Availability

The NHANES II radiographs and images are available through a public access protocol (https://www.cdc.gov/nchs/nhanes/index.htm accessed on 12 September 2022).

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
