# Peer review of "Reference Data for Diagnosis of Spondylolisthesis and Disc Space Narrowing Based on NHANES-II X-rays"

_bioengineering, 2024, doi:10.3390/bioengineering11040360_

Round 1
Reviewer 1 Report
Comments and Suggestions for Authors
1) The radiographic, computed tomographic (CT), and magnetic resonance (MR) imaging findings can help differentiate spinal neuropathic arthropathy from disk space infection. The data of the WebMIRS system consists of major parts of the Second- and Third National Health and Nutrition Examination Surveys (NHANES II and NHANES III). NHANES II was conducted 1976-1980 and included participants aged 6 months to 74 years. The study presents a comprehensive approach toward establishing robust reference data crucial for objectively classifying measurements of spondylolisthesis and disc space narrowing as normal or abnormal. The work outlines the utilization of a large collection of radiographs from the 2nd National Health and Nutrition Examination Survey (NHANES-II) to create reference data. A pipeline involving neural networks and coded logic was employed to place landmarks on vertebrae, aiding in the calculation of multiple disc space metrics. The study concludes with the development of a "SpondyIndex" and highlights the need for caution in interpreting disc metrics, particularly when X-ray images are substantially out-of-plane. The findings of the study underscore the importance of standardized reference data in clinical practice, particularly in the realm of spinal disorders. By leveraging NHANES-II data, the study offers valuable insights into interpreting measurements of sagittal plane offset (SPO), disc heights, and disc angles as normal or abnormal. However, it also acknowledges limitations, such as the need for level-specific reference data and challenges in applying predictions from NHANES data to flexion-extension studies due to low variability in disc angles. The study emphasizes the necessity for further research to determine the clinical efficacy of various metrics and validate diagnosis and treatment algorithms using disc and SPO metrics to enhance outcomes for patients with spinal disorders. It highlights the need for caution in utilizing disc metrics, particularly in cases where X-ray images are not optimally aligned.
2) Additionally, the work includes a rich and extensive review of relevant literature (72). Moreover, supplementary materials accompanying the study further enhance its value, offering additional insights and resources for researchers and clinicians in the field.
3) Comments: Correcting Figure 2 is imperative to maintain the integrity of the work.
4) In my opinion, the article can be accepted for publication.
Author Response
We thank reviewer 1 for the positive comments.
We have removed reference to sagittal plane offset from the caption for figure 2, since the sagittal plane offset is described in figure 3 and not figure 2. We thank the reviewer for pointing out the error.
Reviewer 2 Report
Comments and Suggestions for Authors
This research appears to be warranted based on the literature review. The presentation format is unique in that two of the appendices represent related but distinct studies. This arrangement creates challenges related to the coherence (organization) of the entire report. I have highlighted some of these issues directly on the pdf. The paper is reasonable and well-written, however there are many word tense errors. I have also asked for some clarification on some areas of the paper. See specific comments on the pdf.

Comments on the Quality of English Languagemany word tense errors
Reviewer 3 Report
Comments and Suggestions for Authors
Ref.comments to the paper titled as “Reference data for diagnosis of spondylolisthesis and disc space narrowing based on NHANES-II X-Rays” written by the authors: John Hipp, Trevor Grieco, Patrick Newman, Vikas Patel, and Charles Reitman.
Naturally, for the comfort of human life, it is desirable to be healthy and hardy. In this regard, the condition of the spine, which causes a number of the important functions for human life, is very important to know and maintain. Of course, displacement of the vertebrae, a change in the gap between individual vertebrae can lead to serious consequences. To collect the data about the different technologies, which collect the results about the condition of the spine and individual vertebrae is very important. From this point of view the manuscript is actual and modern.
For the first, the references list is connected with the analysis of 78 scientific publications. Good! But, basically, they are coincided with the old jobs. Please include 5-7 papers written on 2020-2023 years. Some interesting and important phenomena can be found and analyzed as an additionally. It can improve the paper and can help to the readers to compare the results with other ones obtained by different scientific teams in the same area.
The article is well illustrated. The pictures presented are informative and help to understand the material.
Please enter the term designations in your equations. For example, in your Zscore estimation please use the terms k, a, b, etc. Please indicate the number of the equations (1), (2)… as well.
The overall impression of the work consists of a sound statistical analysis conducted for different age groups of the population. It expands our knowledge in the field of the influence of the state of the spine on the quality of life of men and women, and in different age categories. What is a little alarming is that there is no error estimation for each type of statistical analysis. Of course, the risk and limitations are well shown in the paragraph “Limitations of this study include”, but an addition needs to be made on the errors of each analysis method. It is also a little unclear why the appendices have their own numbering of the literary data, and not sequential inclusion in the general body of the article.
Conclusion part should be extended. It is not collect all established tendencies and summary.
I hope that the questions (as the recommendations) are not so complicated for the authors.
Thus, the paper can be published after major corrections.

Round 2
Reviewer 2 Report
Comments and Suggestions for Authors
Thank you for addressing earlier concerns/questions. I have a few additional minor errors (see comments on pdf).
The appendices were taken out of the main manuscript. However, I was able to look at them. It was not evident what was revised. The only thing I saw in a couple of appendices was the use of "std dev" which might be changed to "SD" to be consistent with what was used in the main paper. Also, I am still confused by what exactly is being presented in Table 1 Appendix D. Are the values in the table mean and median values of SD? Also, I am confused by the value in the "95%" column. What does this number represent? I am familiar with a 95% confidence interval, but you present only a single number.

Comments on the Quality of English Languagesome remaining word tense errors that need correction
Author Response
Thank you for the rapid re-review. We had uploaded the appendices as a single zip file per the instructions provided. We are grateful for the effort it took to find them. The revised appendices had been checked for tense errors and these were corrected when found. As an extra check, after manual review, we asked ChatGPT 4: "Are there tense (past versus present) errors in the following text" and then provided text from the appendices. A typical response from ChatGPT ended with: "Overall, the text does not appear to have tense errors, as it employs past tense to discuss completed actions (the experiment and its findings) and present tense for general statements and implications, which is consistent with the conventions of scientific writing."
We spelled out standard deviations the first time we used SD in the manuscript, and consistently used SD to refer to standard deviations throughout the manuscript and appendices.
We reworded the bullet points at the end of the introduction so they are stated as aims.
We checked for and fixed all space errors associated with citing references in the manuscript. There were also some citations that were not captured as Endnote citations and we fixed those.
In section 3.2, we more explicitly describe what we describe as "significant" or "high" correlations.
The equations are now presented to be as consistent as we could achieve with some example recent papers published in the Bioengineering Journal.
Some figure sizes were tweaked to avoid tables from spanning pages.
The word "population" was changed to "sample" when discussing data from a study by Kalichman et al.
The units of degrees was added to the caption for table 3.
Table 1 of Appendix D presents errors in the metrics due to variability in radiographic projection. We describe the upper limit of the errors using the 95th percentile. That is why there is only the one number. We added that clarification.
Reviewer 3 Report
Comments and Suggestions for Authors
Ref.comments to the paper titled as “Reference data for diagnosis of spondylolisthesis and disc space narrowing based on NHANES-II X-Rays” written by the authors: John Hipp, Trevor Grieco, Patrick Newman, Vikas Patel, and Charles Reitman.
This paper has been presented once again after some paragraphs adding to improve the quality.
I have seen the materials of this paper. The “Materials and Methods” section is modified dramatically. “Results” and “Discussion” sections are partially changed. It permits to understand the explanation more adequately.
Conclusion is not expanded. But, this is the vision of the authors, which can be accepted in their concept.
Thus, the paper can be published in the present form.

Author Response
We thank the reviewer for the second review and the positive comments.